# High-Efficiency 4 × 4 × 10 Gbps Orbital Angular Momentum Modes Incorporated into Satellite–Ground–Underwater Optical Wireless System under Diverse Turbulences

**Meet Kumari [1] and Satyendra K. Mishra [2],***

---

[1] Department of Electronics & Communication Engineering, University Institute of Engineering, Chandigarh University, Gharuan, Mohali 140413, India; meetkumari08@yahoo.in
[2] SRCOM, Centre Technologic de Telecomunicacions de Catalunya, 08860 Castelldefels, Barcelona, Spain
* Correspondence: smishra@cttc.es

**Abstract:** With their unique capability to deal with a considerable geographic area, satellite–ground–underwater optical wireless communication (OWC) systems are an appealing alternative to meet the ever-increasing demand for end-to-end broadband services. Using four different Laguerre–Gaussian (LG) modes, an orbital angular momentum (OAM) multiplexing method was developed to enhance the spectral efficiency and system capacity of the satellite–ground–underwater OWC system. At an aggregate throughput of 160 Gbps, LG[0,0], LG[0,2], LG[0,4], and LG[0,8] were realized. Various atmospheric conditions, water types, and scintillation effects were used to evaluate the performance of two separate OWC links for satellite-to-ground and ground-to-underwater communication. A maximum OWC range of 21,500–30,000 km has been obtained under weak-to-strong turbulence for satellite-to-ground scenarios, and a range of 12–27 m underwater for ground-to-underwater scenarios under various scintillation effects. At LG[0,0], in pure sea, the maximum gain is $-75.02$ dB, the noise figure is 75.02 dB, the output signal is $-78.32$ dBm, and the signal-to-noise ratio is 21.67 dB. In comparison with other works in the literature, this system shows a superior performance.

**Keywords:** MDM; OAM; OWC; satellite; underwater

## 1. Introduction

In the coming years, optical wireless communication (OWC) will play an increasingly important role in the development of networks with high capacity and density. There are several advantages to using this technology over radio frequency networks. This technology provides high transmission speeds, requires less installation time, is license-free, secure, has low error rates, and requires little initial investment. Among its applications are trunking networks, connecting buildings, underwater communications, deep space communications, and satellite-to-ground communication [1]. Currently, ground-to-space and ground-to-aircraft communications rely on microwave technology. Eventually, aircraft-to-aircraft links will be OWC. Inter-aircraft optical wireless communication systems can transmit data at speeds of several Gbps over long distances of many kilometers. A satellite-to-ground communication system has been developed utilizing OWC technology [2].

Even though OWC technology has many merits, it also has several disadvantages, including scintillation loss (being sensitive to temperature variations caused by the Earth's heat rise), geometric loss, the attenuation of beam-spreading power, absorption loss (photons absorbed by water molecules or $CO_2$), atmospheric attenuation, and scattering loss [1]. In addition, the ground–underwater communication system can support the development of services like deep-sea mining, high-definition video transmission, and offshore exploration through underwater wireless optical communication (UWOC). It is, therefore, possible to generate high-speed as well as long-distance OWC transmission by using satellite–ground–underwater integrated systems [3].

The mode division multiplexing (MDM) method can be utilized to upgrade the capacity of an OWC link that uses spatial light modes as the information carrier. The orbital angular momentum (OAM) mode has been widely used to improve the capacity of OWC links incorporating the spatial orthogonality of distinct OAM modes [4]. Due to its distinguishability as well as orthogonality through distinct limitless charge numbers, OAM acts as an auxiliary degree of freedom in support of de-/multiplexing and for enhancing overall system capacity [5]. To reduce the complexity of digital signal processing in MDM, the satellite–ground–underwater OWC system is utilized. In this case, the complexity is quantified by the equalizers that are used in MDM reception [6].

Additionally, OWC technology can be deployed efficiently for satellite, ground, and underwater applications. With an OWC transmission scheme, satellite–ground and ground–underwater orbital links can reach long distances with optimum sensitivity to receive signals. An OAM beam offers an increased system capacity as well as spectral efficiency due to its intense level of information carrying capability. In space communications, OWCs are a suitable choice for achieving ultra information transport capacities [7].

## 2. Related Work

In recent years, wavelength division multiplexing (WDM) and OAM have been integrated using multi-mode fibers (MMFs) with a range of 1000 m and a 10 Gbps throughput [4]. In [8], the WDM and OAM techniques were integrated at a 1.6 Tbps data rate. A hybrid OAM, WDM, and orthogonal frequency division multiplexing passive optical network (PON) was designed over 40 km MMF at 40 Gbps throughput in another study [9]. $OAM_{31}$ and $OAM_{41}$ modes are used in [10] to realize an OAM-MDM system over a ring-core fiber range of 300 km and at a 20 Gbps data rate. According to [11], 100 Gbps OAM links incorporating quadrature phase-shift keying multiplexing can provide a 1.5 dB optical signal-to-noise ratio (OSNR) under turbulent conditions. The authors of [12], explored an OAM-based system that offers a 450 Gbps data rate over a 1.5 km range. Ref. [13] also proposed and investigated an OAM-PON system over 0.4 m free space links at 10 Gbps. An MDM system utilizing OAM modes over 1.4 km ring-core fibers at 32 Gbaud is presented in [6]. A 40 Gbps fiber-free space optics (FSOs) system using OAM modes over a 50 km fiber with a 2.5 km FSO range is described in [14].

The authors of [15] further investigated the effect of vehicle motion on a turbulence model by using experimental measurements at a 40 C temperature gradient. The UWOC system in [16] transmits information over a 1.8 m distance in a sea water tank at 3.4 Gbps. Orthogonal frequency division multiplexing over 10 m distance in shallow water is demonstrated in [17]. There is a passive optical network-visible light communication (VLC) integrated system with over 100 km of fiber and a 5 m at 10 Gbps throughput for land–underwater scenarios in [18]. Ref. [3] presents a UWOC system with wavelength division multiplexing over 500 m of free space with a 5 m clear-ocean communication at 100 Gbps. Moreover, ref. [19] presents an integrated FSO and VLC system which can transfer signals over 430 m at 0.96 Gbps and over 1 m at 450 Mbps.

Based on these existing works, this work presents an OAM-based satellite–ground-underwater OWC system that can operate under diverse atmospheric conditions. For satellite–ground–underwater scenarios, four different OAM modes are incorporated. According to our knowledge, this is the first time that higher-order OAM modes have been used for satellite–ground–underwater communication with combined OWC and UWOC links. We investigate the system under the influence of weak-to-strong turbulence, pointing error, geometric loss, and different types of water. These contributions are summarized as follows:

- A high-speed, high-capacity, and long-reach satellite–ground–underwater OAM-based OWC communication system is designed.
- System performance is analyzed for satellite-to-ground and ground-to-underwater communication under diverse climate conditions and different OAM modes.
- System performance is verified w.r.t. another recent works.

The paper is organized as follows: Section 3 depicts the system's design, demonstrating the system through a block diagram, the generated OAM modes, and the system's parameters. Section 4 describes the performance evaluation of the system. A conclusion and the scope of future research are presented in Section 5.

### 3. Proposed Design

Figure 1 depicts the proposed design of the OAM-incorporated satellite–ground–underwater OWC system.

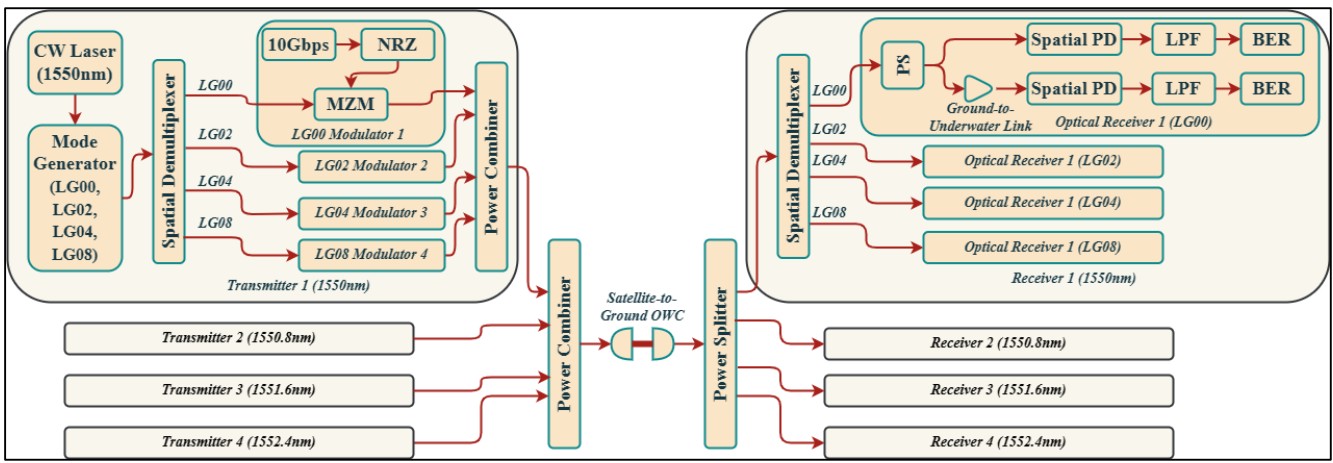

**Figure 1.** Systematic diagram of OAM-incorporated satellite–ground–underwater OWC system.

As depicted, the system is composed of four transmitters/receivers that operate at 1550, 1550.8, 1551.6, and 1552.4 nm wavelengths for satellite–ground–underwater communication. A satellite-to-ground and a ground-to-underwater link is used for satellite-to-ground and ground-to-underwater communication. Four different OAM nodes are incorporated at specific wavelengths in each transmitter. Each transmitter uses a continuous wave laser as an input light source. The mode generator generates four different modes at Laguerre–Gaussian (LG) modes of [0,0], LG[0,2], LG[0,4] and LG[0,8]. In order to multiplex all of these different modes, a spatial multiplexer is used. After that, each incoming mode signal is modulated by a Mach–Zehnder modulator at 10 Gbps without a return to zero. Each transmitter combines these modulated signals using a power combiner. An additional power combiner combines all modulated+mode signals at specific wavelengths for transmission through the OWC link under atmospheric turbulence and link losses. At the receiver, a power splitter splits the received signals into different wavelength sections. In the first step, these signals are demultiplexed via a spatial demultiplexer and then split via a power splitter. Low pas filter (LPF) and bit error rate (BER) analyzer components are used on the first output of the power splitter to convert the optical signal to an electrical one. Using the second output of the power splitter, information is transmitted via UWOC for ground-to-satellite communications. Similarly, a spatial photodetector, and the LPF and BER components are used to obtain the original data. Figure 2 shows the generated OAM modes' two-dimensional (2D) and three-dimensional (3D) views with phase profiles [18].

A multimode generator is used to convert the single-mode signals to multi-mode signals. The multimode generator used in the system attaches LG mode profiles to the input wavelength signals' X and Y polarizations. A Laguerre–Gaussian profile is attached to each polarization. In the proposed system, all spatial modes (LG[0,0], LG[0,2], LG[0,4], and LG[0,8]) are attached to both polarizations (X = Y), where 'LG[]' indicates the mode number starting from fundamental mode at [0,0] to the higher-order mode [0,[8]], taking

both X and Y polarizations as X = Y in the Laguerre–Gaussian beams. Mathematically, the LG mode is defined as [20,21]:

$$\rho_{g,l}(c,\o) = \beta \left(\frac{2c^2}{\omega_0^2}\right)^{\frac{\theta}{2}} . L_g^l \left(\frac{2c^2}{\omega_0^2}\right) . exp\left(\frac{-c^2}{\omega_0^2}\right) . exp\left(\frac{\pi c^2}{\lambda N_0}\right) \begin{cases} \cos(l\o), & l < 0 \\ \sin(l\o), & l \geq 0 \end{cases} \quad (1)$$

where $c$ is the curvature radius, $g$ and $l$ are the x-and y-axis modes' dependencies, $\omega_0$ is spot size, $L_g$ and $L_l$ are the Laguerre polynomials, $N_0$ is the normalized radius, $\theta$ is the beam divergence, and $\beta$ is the atmospheric attenuation coefficient [22].

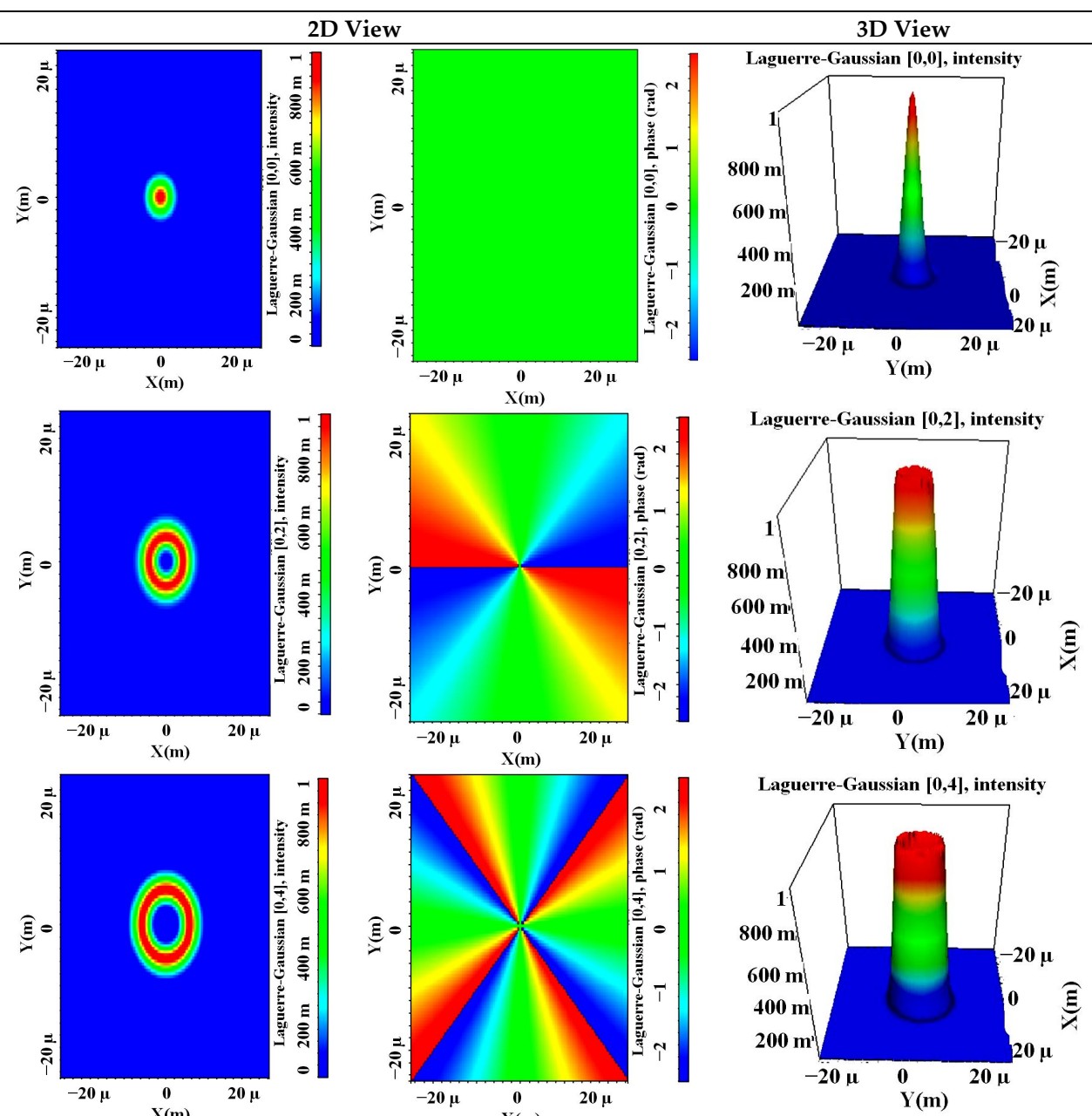

**Figure 2.** *Cont.*

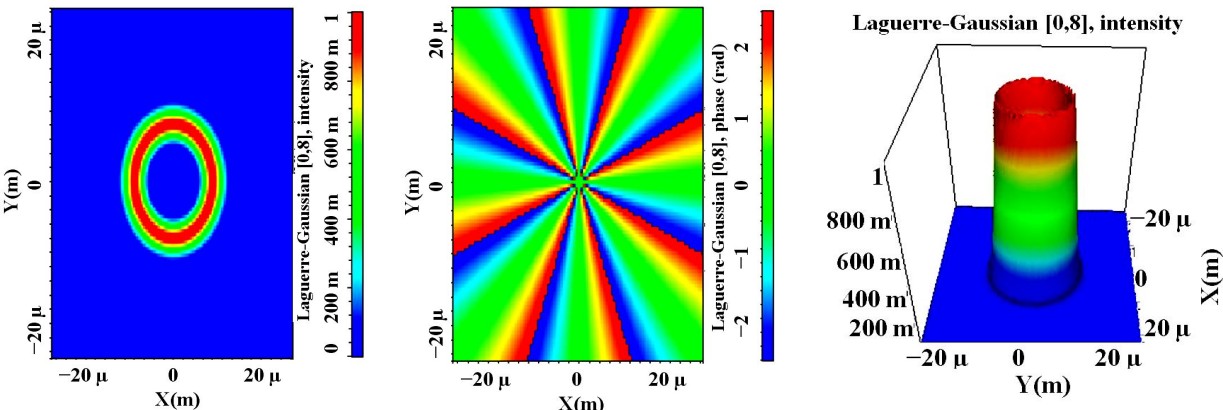

**Figure 2.** Generated OAM modes' 2D and 3D view.

Table 1 depicts various simulation parameters utilized in the proposed design.

**Table 1.** Components' parameters values [18].

| Component | Parameters | Value | Unit |
|---|---|---|---|
| CW laser | Wavelength | 1550, 1550.8, 1551.6, 1552.4 | nm |
| | Power | 0 | dB |
| | Linewidth | 0.1 | MHz |
| | Azimuth | 45 | deg |
| OWC channel | Reference wavelength | 1550 | nm |
| | Range | 21,000–30,000 | km |
| | Tx and Rx aperture diameter | 15 | cm |
| | Tx and Rx optics efficiency | 0.8 | |
| | Free space path loss | Yes | |
| | Geometric gain | Yes | |
| | Tx and Rx pointing error | 0.1 | μrad |
| | Additional losses | 0 | dB |
| UWOC link | Wavelength | 1550, 1550.8, 1551.6, 1552.4 | nm |
| | Range | 5–30 | m |
| | Geometrical loss and gain | Yes | A/W |
| | Scintillation model | Gamma–Gamma | |
| | Aperture diameter | 15 | cm |
| | Beam divergence | 2 | mrad |
| | Transmitter loss | 0.5 | dB |
| | Optics efficiency | 0.9 | |
| Spatial PD | PD | PIN | |
| | Responsivity | 1 | A/W |
| | Dark current | 9 | nA |
| Low pass filter | Cut off frequency | 0.75 × Bit rate | Hz |

### 3.1. Space-to-Ground Atmospheric Model

The Gamma–Gamma channel distribution model used for the proposed design is given as [18,23]:

$$f_{h_a}(h_a) = \frac{2(st)^{\left(\frac{s+t}{2}\right)}}{\Gamma(s)\Gamma(t)}.h_a^{\left(\frac{s+t}{2}\right)-1}K_{s-t}\left[2(sth_a)^{\frac{1}{2}}\right] \tag{2}$$

where $h_a$ means atmospheric turbulence, $\Gamma(.)$ is the Gamma function, $K_{s-t}$ is the modified Bessel function with order $(s-t)$, and $s$ and $t$ are number of large- and small-scale irradiance fluctuations which can be defined as [18,23]:

$$s = \cfrac{1}{\left[ exp\left\{ \cfrac{0.49k_0^2}{\left( 1+0.18d^2+0.56k_0^{\frac{12}{5}} \right)^{\frac{7}{6}}} \right\} - 1 \right]}$$ (3)

and

$$t = \cfrac{1}{\left[ exp\left\{ \cfrac{0.51k_0^2\left( 1+0.18d^2+0.56k_0^{\frac{12}{5}} \right)^{\frac{-5}{6}}}{\left( 1+0.90d^2+0.62d^2k_0^{\frac{12}{5}} \right)^{\frac{5}{6}}} \right\} - 1 \right]}$$ (4)

where $k_0^2$ is the Rytov variance and $d$ is the spherical wave diameter. Again, to evaluate the strength of the atmospheric turbulence, the refraction structure parameter, $C_n^2$, is used, and is defined as (in m$^{-2/3}$) [18,23]:

$$C_n^2 = C_T^2 \times \left( 79 \times \frac{P}{10^6 T^2} \right)^2$$ (5)

where $C_T^2$ is the atmospheric temperature structure parameter, $P$ is atmospheric pressure, and $T$ is the average temperature. Also, the Rytov variance can be used for atmospheric turbulence classification, as [18,23]:

$$k_0^2 = 0.5C_n^2 (2\pi/\lambda)^{\frac{7}{6}} L^{\frac{11}{6}}$$ (6)

where $\lambda$ is operating wavelength and $L$ is transmission length.

### 3.2. Ground-to-Underwater Channel Model

In the proposed design, a line-of-sight (LOS) UWOC link is incorporated from a ground-to-underwater link. The received power for the LOS UWOC is presented as [18,23]:

$$P_{r\_LOS} = P_t\alpha_t\alpha_r exp\left[ -z(\lambda)\frac{d}{\cos(\delta)} \right] \frac{A_R\cos(\delta)}{2\pi d^2[1 - \cos(\delta_0)]}$$ (7)

where $P_t$ is the average power at Tx, $\alpha_t$ is the optical efficiency at Tx, $\alpha_r$ is the optical efficiency at Rx, $d$ is the vertical distance between the Tx and Rx planes, $\delta$ is the angle between the Tx and Rx trajectories as well as that from the normal to the Rx plane, $A_R$ is Rx aperture area, and $\delta_0$ is the laser beam divergence angle. Table 2 indicates the various optical properties of different water types used in the proposed design.

**Table 2.** Optical properties of different water types [18,23].

| Water Type | x($\lambda$) m$^{-1}$ | y($\lambda$) m$^{-1}$ | z($\lambda$) m$^{-1}$ |
|---|---|---|---|
| Pure sea | 0.0405 | 0.0025 | 0.043 |
| Clear ocean | 0.037 | 0.114 | 0.151 |
| Coastal ocean | 0.219 | 0.179 | 0.398 |
| Harbor | 0.913 | 0.187 | 1.1 |

The BER of the proposed design under atmospheric conditions is presented as [18,23]:

$$BER = 0.5\int_0^\infty f_{h_a}(h_a)\, erfc\left( \frac{\langle SNR \rangle s}{2\sqrt{2}\langle i_s \rangle} \right) ds$$ (8)

where $\langle SNR \rangle$ is the mean signal-to-noise ratio (SNR), $i_s$ is mean signal current value, and *erfc* is a complementary error function.

## 4. Results and Discussion

The proposed satellite–ground–underwater OWC system design is demonstrated using the OptiSystem v.21 simulation tool. A BER limit of $10^{-9}$ is used as a threshold limit for performance evaluation. The simulation results are presented for both satellite-to-ground OWC links and ground-to-underwater UWOC links, taking atmospheric conditions into account for different OAM modes. In both the OWC and UWOC channels, Gamma–Gamma distributions are used as they provide a wide range of turbulence from weak to strong. Figure 3a–d depicts 2D intensity profiles of the generated OAM modes viz. LG[0,0], LG[0,2], LG[0,4], and LG[0,8]. Figure 4a–d present the BER performance of the proposed system for varied satellite-to-ground OWC ranges at LG[0,0], LG[0,2], LG[0,4], and LG[0,8] under weak-, moderate-, and strong-turbulence scenarios.

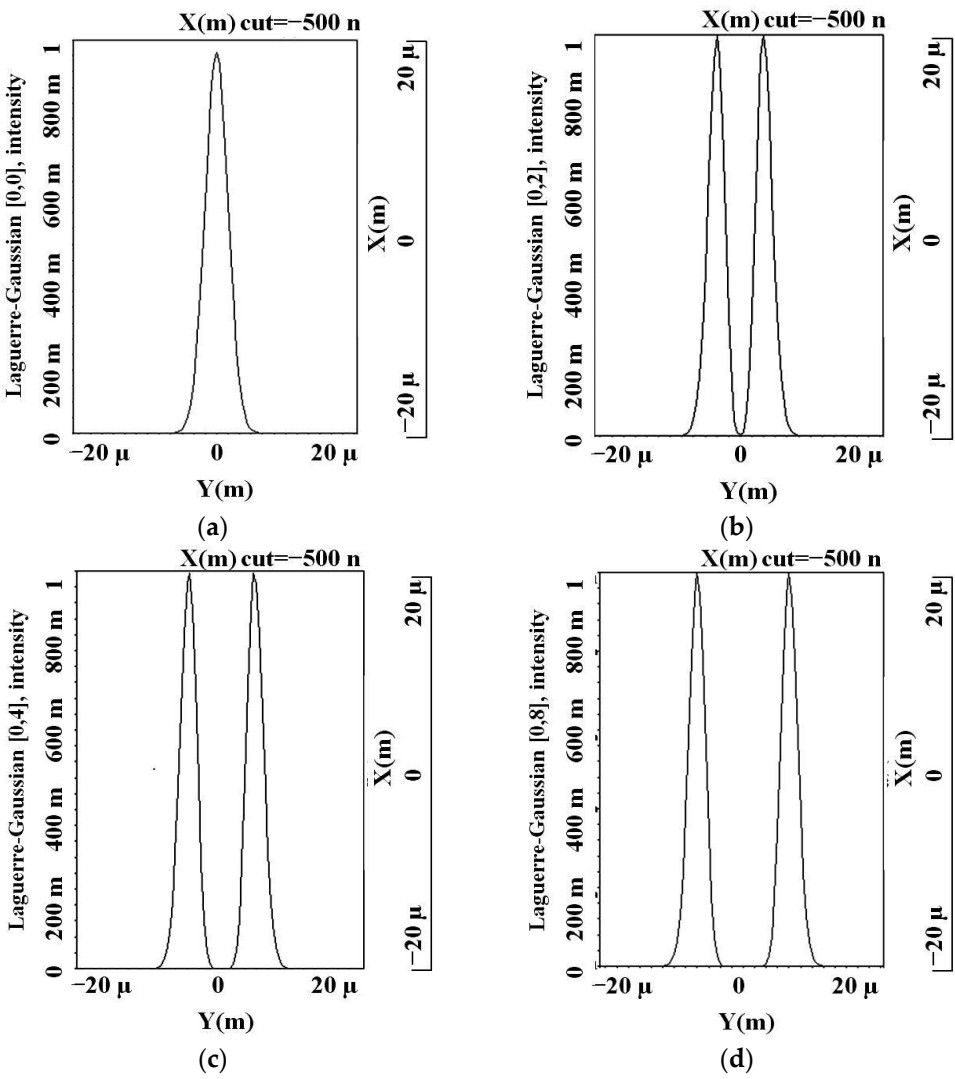

**Figure 3.** Two-dimensional intensity profiles of (**a**) LG[0,0], (**b**) LG[0,2], (**c**) LG[0,4], and (**d**) LG[0,8].

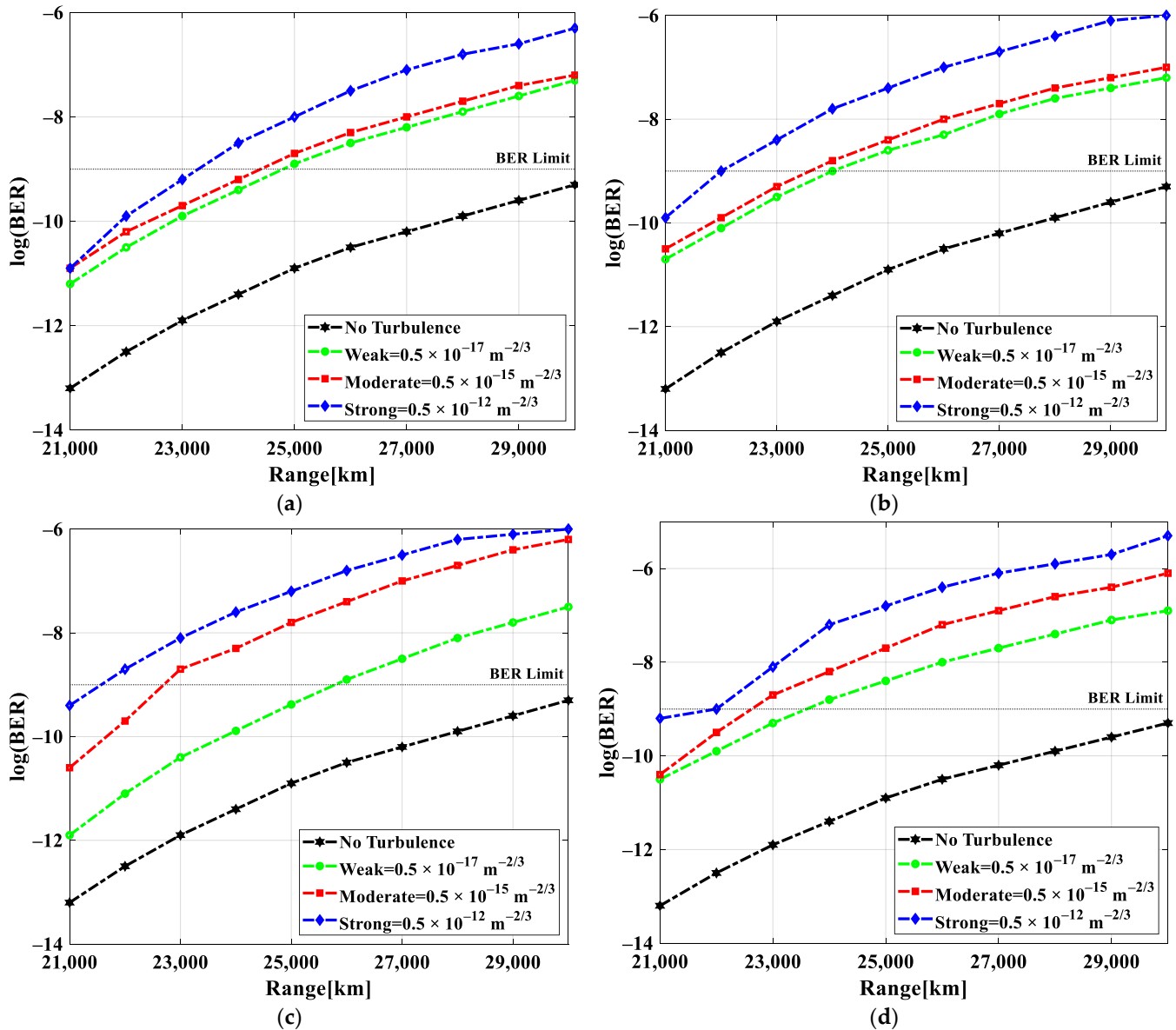

**Figure 4.** BER versus satellite-to-ground OWC range for operating modes of (**a**) LG[0,0], (**b**) LG[0,2], (**c**) LG[0,4], and (**d**) LG[0,8] under diverse turbulent scenarios.

A weak turbulence is defined as one with a refractive structure parameter of $0.5 \times 10^{-17}$, a moderate turbulence as one with a refractive structure parameter of $0.5 \times 10^{-15}$, and a strong turbulence as one with a refractive structure parameter of $0.5 \times 10^{-12}$ m$^{-2/3}$. Here, a BER limit of $10^{-9}$ is considered for system performance evaluation. When the OWC link range is increased from 21,000 to 30,000 km with zero input power at a 1550 nm wavelength, the BER values for all of the modes in diverse turbulent conditions increase. Further, LG[0,0] has a significant increase in transmission range in comparison to any other mode, since it transmits 99% more light than any other mode. Accordingly, in the case of no turbulence, the best performance can be achieved under weak turbulence, followed by moderate turbulence, and the worst under strong turbulence. To evaluate the system's performance under various conditions, the no-turbulence condition has been considered in the simulation as the ideal condition for satellite-to-ground communication over OWC links. Clearly, when the LG[0,0] mode is considered, as shown in Figure 4a, 24,500 km is the maximum range under weak and moderate conditions, while 23,500 km is the maximum range under strong turbulence. For all turbulent conditions, minimum

logarithm (log) BER values of −11 were obtained compared to the no-turbulence conditions which had a log(BER) range from −13 to −9.5. For the LG[0,2] mode, the maximum OWC link ranges were 24,000 km under weak, 23,800 km under moderate, and 22,000 km under strong turbulence, as depicted in Figure 4b. With respect to the no-turbulence conditions, the minimum log(BER) BER obtained was −10.5. The faithful transmission range for LG[0,4] and LG[0,8] modes was, respectively, 26,000 and 24,000 km for weak turbulence; 22,000 and 22,500 km for moderate turbulence; and 21,500 and 22,000 km for strong turbulence (Figure 4c,d). A summary of the maximum OWC link ranges obtained for different modes and turbulent conditions for satellite-to-ground scenarios can be found in Table 3.

**Table 3.** Maximum achieved range for different modes and wavelengths @ BER limit for satellite-to-ground communication.

| Turbulence | LG[0,0] | LG[0,2] | LG[0,4] | LG[0,8] |
|---|---|---|---|---|
| | km | | | |
| No | 30,000 | 30,000 | 30,000 | 30,000 |
| Weak | 24,500 | 24,000 | 26,000 | 24,000 |
| Medium | 24,500 | 23,800 | 22,000 | 22,500 |
| Strong | 23,500 | 22,000 | 21,500 | 22,000 |

Figure 5a–f depict the eye patterns observed at different OWC link ranges for different LG modes under weak turbulence. It can be seen that, with an increase in link range from 21,000 to 30,000 km, distorted eye patters are observed under the no-turbulence conditions, as shown in Figure 5a,b. By considering weak turbulence, a clear and widely opened eye pattern is observed for LG[0,0] over 25,000 km followed by the eye patterns for the LG[0,2], LG[0,4], and LG[0,8] modes, as depicted in Figure 5c–f.

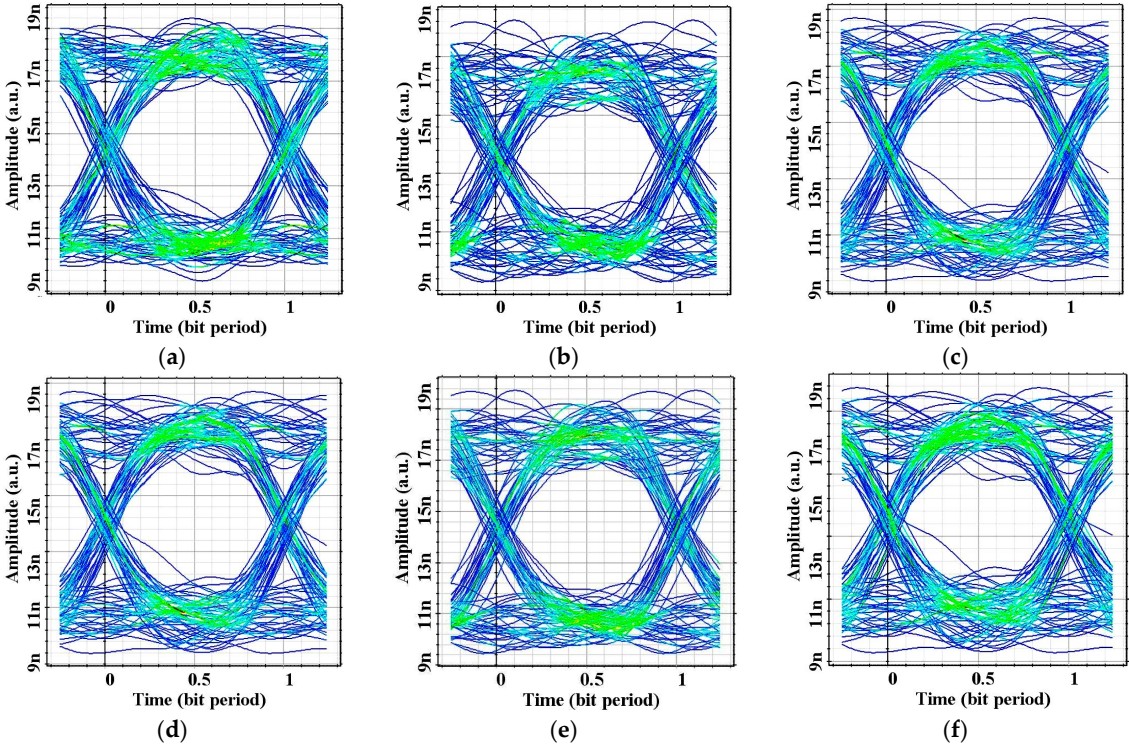

**Figure 5.** Eye diagrams for (**a**) no turbulence at 21000 km, (**b**) no turbulence at 30,000 km, (**c**) weak turbulence over 25,000 km at LG[0,0], (**d**) weak turbulence over 25,000 km at LG[0,[1]], (**e**) weak turbulence over 25,000 km at LG[0,4], and (**f**) weak turbulence over 25,000 km at LG[0,8].

Figure 6a–d show the variation in BER values with the UWOC link range for different turbulences, water types, and operating modes. A maximum transmission range of 50 m is obtained for pure sea with a weak scintillation effect followed by the moderate and then the strong effects. Additionally, clear oceans perform the best, followed by coastal oceans, and harbor waters perform the worst at a log(BER) limit of −9. Moreover, the LG[0,0] mode provides a superior performance in all cases over the other modes. For the LG[0,0] mode, the maximum transmission range is 19–27 m under all scintillation effects and in all types of water, as shown in Figure 6a. Additionally, for the LG[0,2] mode shown in Figure 6b, faithful transmission ranges of 26 m under weak, 20 m under moderate, and 18 m under strong scintillation effects are obtained for all types of water. According to Figure 6c, weak, moderate, and strong scintillation effects offer wireless ranges of 22–24 m, 16–18 m, and 13–15 m, respectively, at LG[0,4]. Figure 6d shows that, for the LG[0,8] mode, the weak, moderate, and strong scintillation effects achieve maximum ranges of 15–23 m, 13–15 m, and 12–14 m, respectively. In Tables 4–6, the results of the proposed system for ground-to-underwater communication under weak-to-strong scintillation effects are summarized for the different modes and water types.

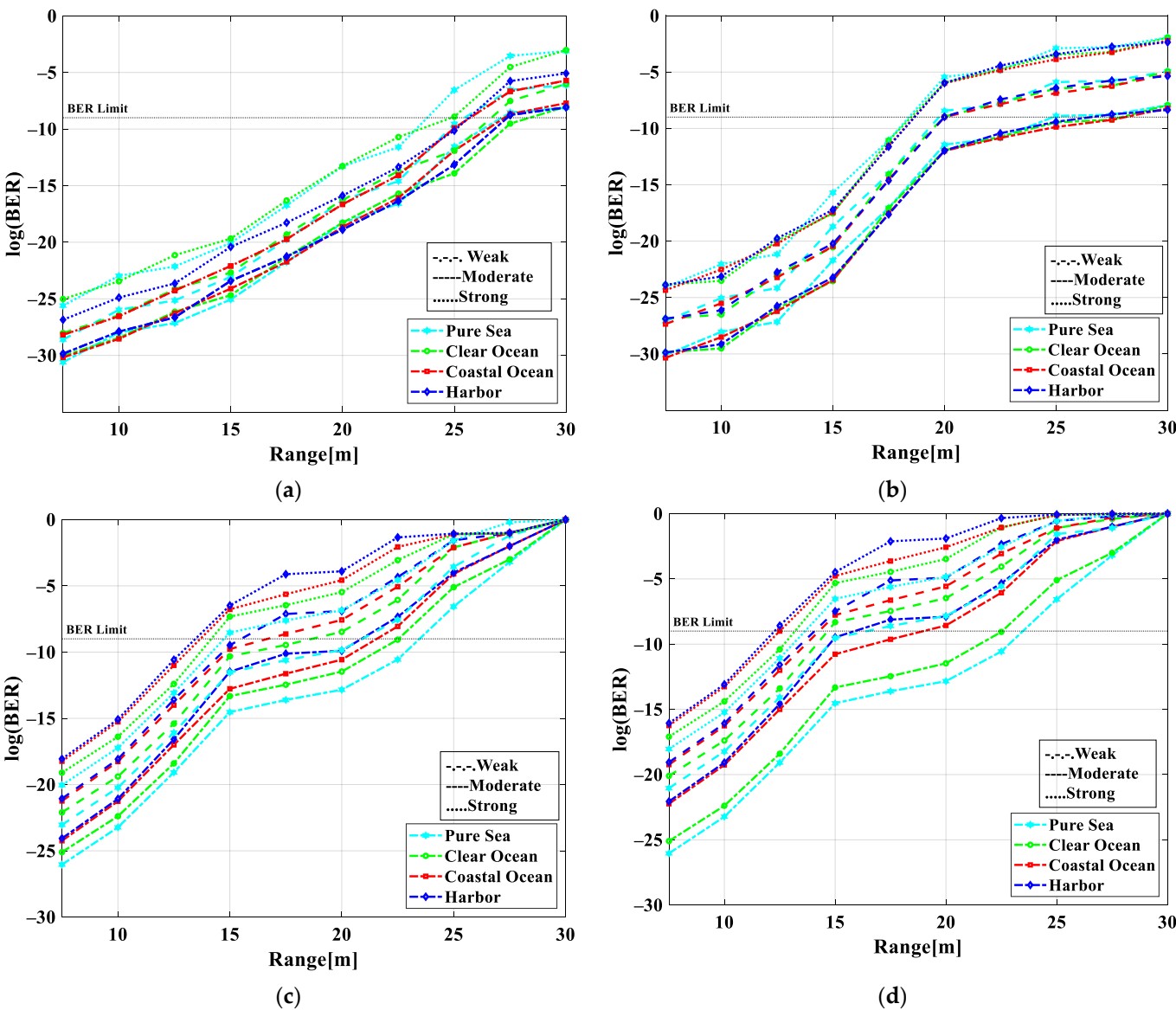

**Figure 6.** BER versus ground-to-underwater transmission range for operating modes of (**a**) LG[0,0], (**b**) LG[0,2], (**c**) LG[0,4], and (**d**) LG[0,8] under diverse scintillation effects.

**Table 4.** Maximum achieved range for different modes and wavelengths @ BER limit for ground-to-underwater communication under weak scintillation.

| Water Type | LG[0,0] | LG[0,2] | LG[0,4] | LG[0,8] |
|---|---|---|---|---|
| | | m | | |
| Pure sea | 27 | 26 | 24 | 23 |
| Clear ocean | 27 | 26 | 24 | 20 |
| Coastal ocean | 27 | 26 | 23 | 17 |
| Harbor | 27 | 26 | 22 | 15 |

**Table 5.** Maximum achieved range for different modes and wavelengths @ BER limit for ground-to-underwater communication under moderate scintillation.

| Water Type | LG[0,0] | LG[0,2] | LG[0,4] | LG[0,8] |
|---|---|---|---|---|
| | | m | | |
| Pure sea | 26 | 20 | 18 | 15 |
| Clear ocean | 26 | 20 | 18 | 15 |
| Coastal ocean | 26 | 20 | 17 | 14 |
| Harbor | 26 | 20 | 16 | 13 |

**Table 6.** Maximum achieved range for different modes and wavelengths @ BER limit for ground-to-underwater communication under strong scintillation.

| Water Type | LG[0,0] | LG[0,2] | LG[0,4] | LG[0,8] |
|---|---|---|---|---|
| | | m | | |
| Pure sea | 25 | 18 | 15 | 14 |
| Clear ocean | 26 | 18 | 15 | 13 |
| Coastal ocean | 25 | 18 | 14 | 12 |
| Harbor | 25 | 18 | 13 | 12 |

Ground-to-underwater optical eye diagrams are shown in Figure 7a–f for distinct distances 5–30 m for pure sea water types. These optical eye diagrams are shown at a 10 Gbps per channel data rate. Eye diagrams are a useful tool for determining system performance, and they are are generated by considering random key bit streams and superimposing them onto each other. These diagrams are observed as a "human eye". These diagrams offer critical information about significant system parameters such as best sampling time, probability of error, rise and fall time, jitter, extinction ratio, and data rate. As depicted in Figure 6a, the eye diagram at the 5 m range illustrates the largest eye opening at the best sampling time with the lowest probability of error. A larger the width of the eye diagram illustrates a lower probability of error occurrence as well as a lower amount of inter symbol interference. Meanwhile, the slopes in the left as well as the right portions of an eye diagram indicate the signal's rise and fall time. A limited rise/fall time depicts a lower bandwidth and thus lower system throughput. Jitter is defined as the width of the lines in the eye diagrams and measured at bits' crossing points. It also helps to identify the error rate in the system. Moreover, the extinction ratio determines the power penalty within the system, whereas the data rate is measured using the inverse of the bit period [24]. Further, it appears that the eye diagrams are wide and clear for shorter distances (up to 20 km) and then become severely distorted after a 20 m distance and become particularly bad after a 20 m distance. Since multiple optical signals are transmitted through a single UWOC link, link distortions such as attenuation loss and geometric loss increase with distance. In water, this results in higher signal distortions over long distances.

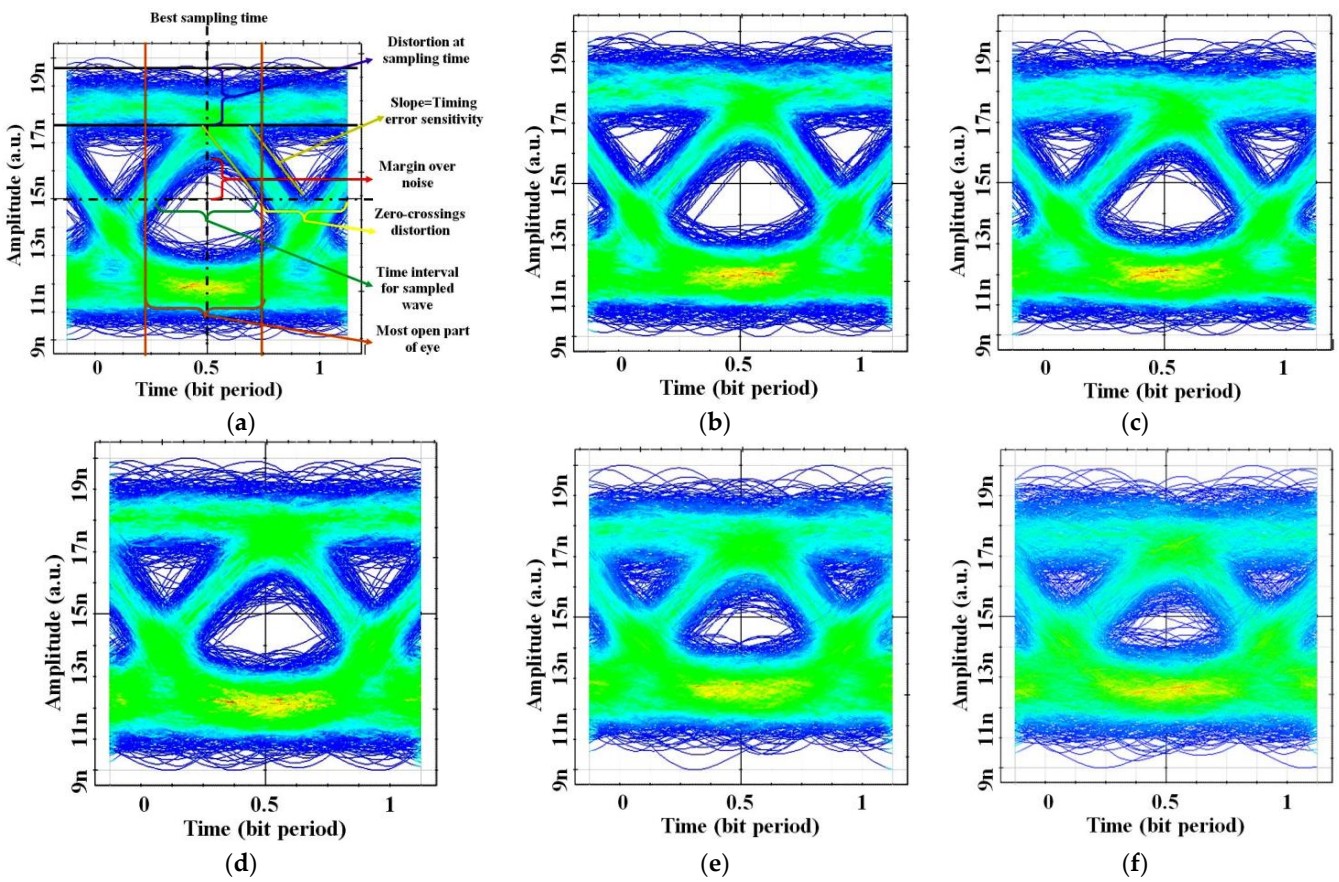

**Figure 7.** Eye diagrams at (**a**) 5 m, (**b**) 10 m, (**c**) 15 m, (**d**) 20 m, (**e**) 25 m, and (**f**) 30 m ground-to-underwater communication for pure sea.

Table 7 shows measured results in terms of gain, noise figure (NF), input-output signal, SNR, and OSNR for the LG[0,0] mode in the weak scintillation scenario within pure sea underwater conditions. Furthermore, Table 8 illustrates the superiority of the proposed design over others.

**Table 7.** Obtained results for LG[0,0] in the weak scintillation scenario.

| Range (m) | Gain (dB) | NF (dB) | Input Signal (dB) | Input Noise (dB) | Input SNR (dB) | Input OSNR (dB) | Output Signal (dB) | Output SNR (dB) | Output OSNR (dB) |
|---|---|---|---|---|---|---|---|---|---|
| 5 | −75.02 | 75.02 | −3.30 | −81.17 | 77.87 | 79.91 | −78.32 | 21.67 | 21.67 |
| 10 | −75.30 | 75.30 | −3.31 | −81.24 | 77.92 | 79.96 | −78.62 | 21.37 | 21.37 |
| 15 | −75.64 | 75.64 | −3.32 | −81.26 | 77.94 | 79.98 | −78.97 | 21.02 | 21.02 |
| 20 | −75.95 | 75.95 | −3.30 | −81.20 | 77.90 | 79.94 | −79.25 | 20.74 | 20.74 |
| 25 | −76.22 | 76.22 | −3.31 | −81.23 | 77.91 | 79.95 | −79.54 | 20.45 | 20.45 |
| 30 | −76.44 | 76.44 | −3.30 | −81.19 | 77.88 | 79.92 | −79.75 | 20.24 | 20.24 |
| 35 | −76.77 | 76.77 | −3.32 | −81.28 | 77.95 | 79.99 | −80.10 | 19.89 | 19.89 |
| 40 | −76.99 | 76.99 | −3.31 | −81.31 | 77.99 | 80.03 | −80.31 | 19.68 | 19.68 |
| 45 | −77.29 | 77.29 | −3.31 | −81.15 | 77.83 | 79.87 | −80.60 | 19.39 | 19.39 |
| 50 | −77.53 | 77.53 | −3.31 | −81.27 | 77.96 | 80.00 | −80.84 | 19.15 | 19.15 |

**Table 8.** Comparison of performance w.r.t. existing works.

| Ref. | No. of Modes | Mode | Data Rate (Gbps) | Wireless Range (m) | Turbulent Condition | Underwater Link | No. of Chansnels | SNR (dB) | BER |
|---|---|---|---|---|---|---|---|---|---|
| [25] | Not used | Not used | 0.0126 | 500 k | Weak, moderate, and strong | Not used | 2 | Not defined | $10^{-3}$ |
| [26] | 2 | 00, 01 | 20 | 1750 k | Not defined | Not used | 6 | Not defined | $10^{-9}$ |
| [27] | 4 | 01, 02, 03, 04 | 2.488 | Not used | Not used | Not used | 4 | Not defined | $10^{-9}$ |
| [23] | Not used | Not used | Not defined | 1.5 k | Weak–strong | Not used | Not defined | 36 | $10^{-3}$ |
| [28] | Not used | Not used | 0.622 | 500 k | Moderate | Not used | 3 | 22 | $10^{-9}$ |
| [29] | 9 | 00, 01, 02, 10, 11, 12, 20, 22, 21 | 10 | 3200 k | Not defined | Not used | 10 | Not defined | $10^{-9}$ |
| [30] | Not used | Not used | 10 | 160 k | Not used | Not used | Not defined | Not defined | $10^{-9}$ |
| [31] | 3 | 01, 02, 03 | 1.866 | Not used | Not used | Not used | 3 | Not defined | $10^{-9}$ |
| This work | 4 | 0,0; 0,2; 0,4; 0,8 | 160 | 36,000 k + 50 | Weak to strong | Yes (Pure sea, clear ocean, costal ocean, harbor) | 4 | 21.67 | $10^{-3}$ |

## 5. Conclusions

This paper designs and investigates a satellite–ground–underwater OWC system based on OAM technology with $4 \times 4 \times 10$ Gbps transmitting nodes. Based on the results of the analysis, it can be concluded that OAM using different LG modes offers significant benefits in all three scenarios within space, air, and the ocean for enhancing the range and quality of transmissions, the data rate, and the channel capacity [32]. With a BER of $10^{-9}$, a maximum satellite-to-ground transmission distance of 21,500–30,000 km can be achieved. Considering weak, moderate, and strong scintillation effects, a maximum transmission distance of 12–27 m can also be achieved for ground-to-underwater communication in pure sea, clear ocean, coastal ocean, and harbor water. Additionally, the system provides a high gain, NF, received signal, and SNR in 5 m underwater communications of $-75.02$ dB, 75.02 dB, $-78.32$ dBm, and 21.67 dB, respectively. This proposed satellite–ground–underwater OWC system demonstrates wide and open eye patterns for an underwater range of 5–30 m. Additionally, this design offers an optimum performance compared to other existing designs. It will be possible to implement this system in the future for satellite-to-ground transmissions with high bandwidths and long ranges, and vice versa.

**Author Contributions:** M.K. and S.K.M. discussed the plan and agreed on it. M.K., drafted designs for the manuscript. The manuscript original was written by M.K. Manuscript edited by S.K.M. They all reviewed and commented on the original draft of the manuscript. All authors have read and agreed to the published version of the manuscript.

**Funding:** S.K.M wants to thank CTTC for providing the resources to conduct this research.

**Institutional Review Board Statement:** Not applicable.

**Informed Consent Statement:** Not applicable.

**Data Availability Statement:** Data is contained within the article.

**Conflicts of Interest:** The authors declare no conflicts of interest.

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
