# Peer review of "High-Efficiency 4 × 4 × 10 Gbps Orbital Angular Momentum Modes Incorporated into Satellite–Ground–Underwater Optical Wireless System under Diverse Turbulences"

_photonics, doi:10.3390/photonics11040355_

Round 1

Reviewer 1 Report

Comments and Suggestions for Authors

It is a good work; however, there are some areas that can be improved.

* We know that a BER result of 10e-9 is generally considered an acceptable BER for telecommunication whereas 10e-13 is more appropriate minimum BER for data transmission. So, I would like to see plots for 10e-9 BER data in this paper. 

* And, from eye diagrams, what are the parameters to be extracted to claim that the eyes are acceptable, should be explained in the paper. 

* Authors claimed about 30,000,000m to 36,000,000m satellite communication. Is it correct data? Can the author include some eye diagrams for this distance?

 * Authors claimed 160G data but it is not clear what speed data in their eye diagrams. Fig. 6 should include the speed information.

Author Response

Attached the response 

Reviewer 2 Report

Comments and Suggestions for Authors

Using four different Laguerre- 12 Gaussian (LG) modes, an orbital angular momentum (OAM) multiplexing method was developed 13 to enhance the spectral efficiency and system capacity of the satellite-ground-underwater OWC sys- 14 tem. At an aggregate throughput of 160Gbps, LG [0,0], LG[0,2], LG[0,4] and LG[0,8] are realized. 15 Various atmospheric conditions, water types, and scintillation effects are used to evaluate the per- 16 formance of two separate OWC links for satellite-to-ground and ground-to-underwater communi- 17 cation. This manuscript is interesting. It can be accepted in this state.

There is only one question. How about the measured results of this scheme?

Comments on the Quality of English Language

The english is good.

Author Response

Attached the response

Reviewer 3 Report

Comments and Suggestions for Authors

The authors demonstrated the design of OAM incorporated satellite-ground-underwater OWC system and investigate corresponding performances in theory. This work may be interesting for the community of optical communication and can be accepted after addressing follow comments.

1. What are the LG [0,0], LG [0,2], LG [0,4], LG [0,8]? They definition of LG [] should be clarified.

2. The mode modulation and demodulation are vague. Corresponding optical methods and  technologies should be presented.

3. The figures are confused. In Figure 2, the color bars are missed and what is the z axis of 3D view meaning, 200m-800m its distance? or intensity? The same problem also show in Figure 3.

4. The definition and scale of axis in Figure 6 are missed

Author Response

Attached the response

Round 2

Reviewer 1 Report

Comments and Suggestions for Authors

Author should revise the Abstract and the conclusion to reflect the revised/new results. 

Author Response

Attached the response 
